# Measuring Quality of Life in Parents or Caregivers of Children and Adolescents with Celiac Disease: Development and Content Validation of the Questionnaire

**DOI:** 10.3390/nu11102302

**Published:** 2019-09-27

**Authors:** Liliane Maria Abreu Paiva, Lenora Gandolfi, Riccardo Pratesi, Rosa Harumi Uenishi, Renata Puppin Zandonadi, Eduardo Yoshio Nakano, Claudia B. Pratesi

**Affiliations:** 1Interdisciplinary Laboratory of Biosciences and Celiac Disease Research Center, School of Medicine, University of Brasilia, Brasilia 70910-900, DF, Brazillenoragandolfi1@gmail.com (L.G.); pratesiunb@gmail.com (R.P.); rosa.uenishi@gmail.com (R.H.U.); 2Department of Nutrition, School of Health Sciences, University of Brasilia, Brasilia 70910-900, DF, Brazil; renatapz@yahoo.com.br; 3Department of Statistics, University of Brasilia, Brasilia 70910-900, DF, Brazil; eynakano@gmail.com

**Keywords:** quality of life, celiac disease, parents, caregivers

## Abstract

Celiac disease (CD) is an autoimmune disorder triggered by the ingestion of gluten and affects approximately 1% of the global population. Currently, the only treatment available is lifelong strict adherence to a gluten-free diet (GFD). Chronic diseases such as CD affect patients and their family members’ quality of life (QoL); particularly parents and caregivers who play an essential role in the child’s care and treatment. A higher level of psychological distress has been found in the parents of children with chronic ailments due to limited control over the child’s daily activities and the child’s illness. In this context, the validation of a specific questionnaire of QoL is a valuable tool to evaluate the difficulties faced by parents or caregivers of children with this chronic illness. A specific questionnaire for this population can elucidate the reasons for stress in their daily lives as well as the physical, mental, emotional, and social impact caused by CD. Therefore, this study aimed to develop and validate a specific questionnaire to evaluate the QoL of parents and caregivers of children and adolescents with CD. Overall results showed that a higher family income resulted in a higher score of the worries domain. In addition, having another illness besides CD decreased the QoL (except in the worries domain). The other variables studied did not present a statistically significant impact on the QoL, which was shown to be low in all aspects. Knowledge of the QoL is important to help implement effective strategies to improve celiac patients’ quality of life and reduce their physical, emotional, and social burden.

## 1. Introduction

Celiac disease (CD) is a common and lifelong autoimmune condition that can occur at any age and is caused by an abnormal reaction to the ingestion of gluten (a protein found in wheat, barley, and rye) where the immune system reacts by damaging the lining of the small intestine [1,2]. CD affects about 1% of the global population [1,3,4,5,6]. Currently, the only treatment available is lifelong strict adherence to a gluten-free diet (GFD). Therefore, once diagnosed, it is necessary for CD patients and family members to adapt to a strict GFD and the logistics of buying, processing, and storing gluten-free (GF) meals [7,8]. Adherence to a GFD is difficult because of the emotional, economic, and social challenges associated with this dietary limitation [7,9,10]. Patients tend to transgress on their diet for several reasons: the high cost of GF products, lack of appropriate dietary guidance, lack of cooking skills, disbelief concerning the number of prohibited products, and long-rooted habits of consuming foods that contain grains such as wheat, rye, and barley [11]. It is important to mention that the safety of oats for consumption by celiac patients has been the subject of controversy and the immunogenicity of oats varies, depending on the cultivar consumed and the level of gluten-contamination [12]. In some cases, avoiding the consumption of oats is recommended [12]. Compliance with a strict and lifelong GFD requires continual effort and attention from patients and, in the case of children and adolescents, from their parents or caregivers [8,11].

Family involvement is inevitable in CD treatment since family members need to take on the responsibility for dietary adherence. In general, although a GFD may not be fully adopted by unaffected family members, it will at the very least influence social events and meals consumed at home. Family involvement is not limited only to the support in dietary adherence; it is also the support needed to deal with the effect that the diagnosis has on patients [13]. Chronic diseases such as CD affect patients and their family member’s quality of life (QoL); particularly those parents who play an essential role in the child’s care and treatment. A higher level of psychological distress has been found in several clinical observations, where parents of children with chronic ailments have limited control over the child’s daily activities and anxiety due to the child’s illness [14,15]. The burden of parents or caregivers can culminate in acute and chronic physical disorders, resulting in isolation and depression as well as financial imbalance, and self-blame [16]. The restrictive nature of a GFD has a negative impact on the QoL of celiac patients and their families, especially parents or caregivers, which in turn may make adhering to the required diet difficult [17,18].

Even though a GFD can bring relief from the physical symptoms of CD, however, as mentioned, adherence to a GFD is a challenge for many patients. The non-adherence or support for a GFD by family members and social groups may compromise activities such as vacations, trips, family events, or restaurant outings for the patient, therefore, impacting the well-being and the QoL of the CD patients and consequently their families [8,13,14,19,20]. Questionnaires assessing the QoL are a valuable tool and are increasingly used by health professionals to evaluate the problems and difficulties caused by CD and the QoL of patients on a GFD [9,10,21,22]. A study [20] aimed to explore the dilemmas experienced by close relatives living with children with CD. The authors interviewed twenty-three family members, which revealed that the CD-related worries included having a bad conscience about not being affected by the disease, experiencing anxiety, and witnessing the vulnerability of the affected individual in social situations. Managing a strict GFD is challenging because it frequently involves an increase in cooking meals at home and avoiding social events that interfere directly with the family’s social life and [20] potentially impacts their QoL.

To our knowledge, there have only been two studies focused on the QoL of celiac parents and caregivers [8,16]. A study conducted in Brazil aimed to evaluate the QoL of parents (*n =* 63) of celiac children [8] using a generic questionnaire (WHOQoL-BREF). A study conducted in Turkey with mothers of celiac patients (*n =* 40) aimed to identify traumatic symptoms in mothers and evaluate their QoL by utilizing the generic questionnaires “Post Traumatic Stress Disorder Checklist-Civilian Version (PCL-C)” and “Short Form-36 (SF-36) [16].” To our knowledge, this is the first QoL questionnaire developed and validated to specifically explore the impact of CD on parents or caregivers of celiac children and adolescents.

This study aimed to develop and validate a specific questionnaire to evaluate the QoL of parents or caregivers of children and adolescents with CD. It aimed to assess the financial, physical, mental, and emotional concerns of the parents and caregivers. The knowledge of what it means to live with a person with CD from the perspective of parents can be useful for healthcare personnel in the support they give to relatives, and could thus increase the chronically ill person’s opportunities to obtain support from their network. It can also be a tool to prevent psychological or social complications of the disease for the individual and their family [19,20]. We hope that the present study will help health professionals and government institutions develop effective strategies to improve the QoL of parents or caregivers of celiac patients, favoring the treatment of children and adolescents.

## 2. Materials and Methods

The study was developed in six steps: (i) development of the CD parent/caregiver QoL questionnaire (CDPC-QOL); (ii) subjective evaluation; (iii) validation of the questionnaire by Delphi method; (iv) evaluation of the internal consistency and reproducibility of the CDPC-QoL; (v) application of the questionnaire to Brazilian celiac parents or caregivers; and (vi) statistical analysis. The study was approved by the Health Sciences Ethics Committee, University of Brasilia, number 01029018100000030, and followed the guidelines established by the Declaration of Helsinki.

### 2.1. Development of the Questionnaire

The questionnaire was formulated based on extensive literature review and the researcher’s experience on the matter. Additionally, general QoL questionnaires were used such as the CD-QoL [9,10,23]; and questionnaires designed to evaluate the QoL of parents of children with other chronic diseases such as type 1 diabetes or cancer were also used [15,24,25]. Topics and items from previous studies were carefully assess, and those thought to be relevant to evaluate the QoL of celiac children and adolescent’s parents were chosen and adapted for the initial version of the questionnaire. Similar to other studies of CD-QoL, we adopted three domains: Emotional, Worries, and Social [9,10,26,27], where the possible physical, mental, emotional, and social aspects that may compromise activities such as vacations, trips, family events, or restaurant outings for the patient, therefore, impacting on well-being, caused by CD in the daily lives of parents and caregivers.

### 2.2. Subjective Evaluation

For the subjective evaluation, experts with known experience in the treatment of CD were invited to participate. Of those invited to participate in the panel, 13 agreed to participate in the study of which four were pediatricians (three of them with specialization in gastroenterology), six gastroenterologists, one psychologist; one dietitian (with expertise in gluten-related disorders), and one dentist who worked with celiac patients and their families at the University Hospital of Brasilia CD outpatient clinic. According to Pasquali [28], a minimum of six judges with expertise in the subject area are necessary to compose a panel of the specialists. The experts received the information and guidance needed on the Delphi method of evaluation.

In the first phase of the Delphi method, the expert panel was asked to evaluate the initial 48 questions developed. They were asked to express their opinion on the preliminary version of the instrument and evaluate the overall questionnaire, considering aspects such as the content, clarity, type, and consistency of the items. Experts were also asked to suggest any modification, exclusion, or inclusion of items they judged relevant and to freely comment on any subject regarding the questionnaire, characterizing the qualitative analysis stage.

### 2.3. Semantic Evaluation and Content Validation

The instrument validation consists of a methodological procedure to evaluate its quality, regarding the capacity of the instrument to accurately measure what it is intended to measure [29,30]. Therefore, the validation of the CDPCA-QoL occurred in two different steps. In the first step, the semantic evaluation and content validation were analyzed by a panel composed of professionals and researchers recognized in the CD treatment (as mentioned in item 2.2). An expert panel consensus defines the instrument items that should be maintained, revised, or excluded [31,32].

We used the Delphi method for content validation. This method utilizes the anonymous response of experts to achieve a consensus on a specific subject in situations where new ideas are created [33]. It is a method, in which, through collegial communication ordered by individual responses often conducted by questionnaires, the consensus of a group is achieved [33].

The MonkeySurvey^@^ platform was used for the content validation of the CDPCA-QoL. The first page of the CDPCA-QoL explained the evaluation criteria for the questionnaire. Experts were informed to evaluate each item using a Likert scale ranging from (1) “I fully disagree with the item” to (5) “I fully agree with the item”, and were asked if the item should be maintained or not in the questionnaire. Once all experts gave their feedback, the questionnaire was analyzed, and items were either approved, modified, or deleted according to suggestions made by the experts [30]. Once modifications were made, the new version of the questionnaire was sent to the group of specialists for an additional round. This procedure was used to obtain a consensus among the experts.

### 2.4. Reproducibility Analysis

The reproducibility of the CDPC-QoL was evaluated using 10 CD parent’s responses. The CD parents answered the CDPC-QoL, and one week later they were invited to answer the CDPC-QoL again. The test–retest reliability (reproducibility) of the questionnaire was verified by the intraclass correlation coefficient (ICC).

### 2.5. Brazilian Questionnaire (CDPCA-QoL) Application

The final step was to place the *CDPCA-QoL* questionnaire on the MonkeySurvey^®^ platform and applying it to Brazilian CD parents or caregivers to measure their QoL. The first page of the survey presented the consent form, which included the established exclusion/inclusion criteria. At that point, participants gave their consent. Individuals that did not agree to participate were directed to a page thanking them for their time; while those that agreed were directed to the first page of the survey.

The inclusion criteria for those who agreed to participate in the research; the children or adolescent should be following the GFD (based on a self-reported question). Parents were excluded from the study if their children had not been diagnosed by a physician, had less than a year of diagnosis, or was over 18 years of age or under 12 months. The cut-off point for “under 12 months” was used based on the inclusion criteria, where the children were required to have at least a year of diagnosis. In Brazil, the Brazilian Society of Pediatrics [34] recommends exclusive breastfeeding until six months, and the introduction of gluten after six months of age.

We used the Brazilian Celiac Association (ACELBRA) to help us distribute the questionnaire among their associates. We also used social media to reach out to the participants.

### 2.6. Statistical Analysis

A confirmatory factor analysis was used to assess the factor validity. The root mean square error of approximation (RMSEA) and the Chi-squared (χ^2^) test of minimum discrepancy evaluated the factor validity. The RMSEA ranged from 0 to 1, with smaller values indicating a better model fit. A value of 0.05 or less is indicative of acceptable model fit. The internal consistency of the CDPC-QoL and its three domains was verified through Cronbach’s alpha measure. The QoL scores were described in terms of mean and standard deviation (SD). Comparison of the CDPC-QoL scores and their domains was performed by the Student’s *t*-test (for variables with two categories) and by analysis of variance (ANOVA) followed by Tukey’s posthoc tests (for variables with three or more categories). All tests were performed considering bilateral hypotheses and a significance level of 5%. The analyses were performed by the SPSS (Statistical Package for Social Sciences) version 22 and SPSS AMOS (analysis of moment structures) Version 20.0.0.

In the final version of the questionnaire, most (70%) of the questions (1–4; 6; 8–10; 13; 15–17; 20–23; 25; 26; 28; 30) that remained in this study were adapted from the original version of the CD-QoL [10]. We used this questionnaire due to its capacity to evaluate the attitudes and perceptions of celiac patients, covering social, worries, and emotional aspects. From the CD-QoL, we used the three domains: (i) emotions (depressed, restless, relaxed, happy, physically fatigued, tearful); (ii) worries (being diagnosed too late, fear of medical examinations, afraid of cancer, lack of medical expertise, problems with health providers, inheritance of the disease to children); and (iii) social (lack of understanding by colleagues, difficulties in recreation/sports, professional limitations, lack of understanding by family/friends, invitation/dinner, feeling of exclusion from others, sexual activities). Other questions (20%) that remained in the final version (7, 11, 14, 24, 27, 29) were adapted from the Caregiver Burden Questionnaire for Heart Failure (CBQ-HF) [25]. Moreover, questions 12, 18, and 19 were inserted by the panel of specialists (10%), based in their expertise, and approved in the final version.

## 3. Results

### 3.1. Development of the Questionnaire

The initial questions, 48 in total, were elaborated for the questionnaire based on an extensive literature review and considering the suggestions made by the experts such as nutritionists, pediatricians, and gastroenterologists with experience with CD. Once the questions were created, these were divided into three domains: emotion, worries, and social. Figure 1 summarizes the stages of the Brazilian questionnaire process.

Experts of the interdisciplinary CD laboratory performed the subjective evaluation and based on their suggestions, 18 items were excluded, three were included, and nine were modified. A total of 33 questions remained.

Fourteen judges were invited to participate in the objective evaluation (semantic evaluation and the content validation), and 13 agreed to participate. In the content validation, three rounds were necessary to obtain an agreement among the experts. In the first round of objective evaluation, 29 of the questions were approved with an 80% or over approval rate, and were considered adequate regarding reliability, clarity, and easy comprehension. The experts suggested changes to the questions that were not approved, and only the modified questions (*n =* 4) were sent for a second round for evaluation. In the second round, we obtained answers from 12 experts, where 30 questions were approved and three were excluded. Therefore, the questions obtained 100% of approval at the end of the third round. The final version of the questionnaire presented 30 items (Appendix A).

The questionnaire was composed of 30 items with answers on a five-point Likert scale from 1 to 5. The value of the index was defined as the sum of the answers for each of these items. Thus, we can assume values between 30 and 150 for the *CDPCA-QoL* questionnaire. The higher the QoL value, the greater the QoL. This index was subdivided into three domains (emotions, worries, and social) with 10 items each. Each of these domains can assume values between 10 and 50. The higher the value of the score, the higher the QOL within the domain.

### 3.2. Factor Validity, Reproducibility, Responsiveness, and Internal Consistency of the Brazilian CDPCA-QoL Questionnaire

The internal consistency (reliability) of the instrument (and its domains) was verified by Cronbach’s alpha measure (Table 1). The questionnaire also presented good internal consistency (alpha = 0.913; 95% CI: 0.892–0.932) for all domains (alpha > 0.8). The intraclass correlation coefficient (ICC) was used to verify the reproducibility of the questionnaire. The CDPC-QoL questionnaire presented appropriate reproducibility (ICC = 0.88; *p* < 0.001) and internal consistency (reliability). In addition, the responsiveness of the questionnaire was verified by floor and ceiling effects, and the results obtained showed a good responsiveness (floor and ceiling effects ≤2.7%) of the CDPC-QoL [35]. Factor validity was examined by confirmatory factor analysis. The three domains presented appropriate fit in the confirmatory factor analysis: RMSEA = 0.022 (95% CI: 0.000-0.038) and χ^2^ = 367.8; df = 344; *p* = 0.181.

### 3.3. CDPCA-QoL Questionnaire Application

From January through to April of 2018, a link to the CDPC-QoL questionnaire was distributed nationwide to multiple Brazilian Celiac Associations by email. The associations emailed the link to parents of children and adolescents with CD that were registered with them. The link to the questionnaire was also distributed through CD support groups and subsequently shared by members. Additionally, dietitians and gastroenterologists were also asked to distribute the link to the parents of their CD patients. Therefore, a convenience sample was used to perform the present study.

A total of 150 participants agreed to participate and thoroughly answered the specific parts of the QoL questionnaire. The individuals took an average of six minutes to complete the questionnaire. Table 2 presents the characteristics of the respondents and their association with the QoL subcategories. All respondents completed the QoL questionnaire, however, missing answers were observed in several study variables. Table 2 presents the total number of responses of each variable. The marital status was divided into either “with a partner” (married or with a live-in partner) or “without a partner” (those that are single, divorced, or widowed). Most of the respondents were classified as “with a partner”; however, the marital status did not present significant effect in QoL.

Most of the participants (89%, *n =* 132) were female. However, the QoL did not differ considering the gender of the respondent. The results showed that the higher the family income, the lower the worry (higher score of the domain worries) (Table 2). Additionally, having another illness besides CD decreased the QoL (except in the worries domain). The other variables studied did not present a statistically significant impact on the QoL.

Eighteen percent (*n =* 27) of the participants used antidepressant or anti-anxiety medication, however, we did not find statistical differences in QoL among the group that was taking or did not use these medications. Almost 84% (*n =* 114) of the parents mentioned that their child experienced discomfort or CD symptoms with gluten consumption, however, regardless of the presence or absence of symptoms with gluten consumption, the QoL of the celiac parents was affected by their children’s disease. It is noteworthy that regardless of the age of their child, the QoL was also equally affected. Only 14% (*n =* 22) of the parents who answered the questionnaire were also diagnosed with CD. However, the QoL of parents or caregivers who had CD and those that were not diagnosed with CD did not differ.

It is important to mention that the answers to two of the sociodemographic questions, items 15 and 16, were not presented in Table 2 since item 15 “Do you prepare GF meals at home?”, and 16 “Was the CD of your child or adolescent diagnosed by a doctor?” did not present variability in the responses (absolute agreement was 95%), therefore, those were not considered statistically significant (*p* < 0.05). It is also important to highlight that our sample size was relatively small, 150 individuals, and the variables that presented as < 150 indicate missing sociodemographic data, where the participant did not mark any response in the item of the questionnaire.

## 4. Discussion

According to the World Health Organization [36], health is defined as physical, mental, and social wellbeing and to achieve optimal health, it is essential to comprehend the individual’s perception of QoL [36,37]. The long-term consequences of childhood or adolescent CD can negatively impact the health of the parents or caregivers because they experience greater stress than parents of healthy children or adolescents [24,38,39]. There is vast scientific literature on the impact of diagnosis and treatment in children or adolescents and the impact on their QoL [8,14,16,23,40,41,42]. However, the literature on parental or caregiver outcome associated with CD treatment is limited [8,16].

Therefore, it is necessary to comprehend the QoL of parents and caregivers to help them achieve optimal health [37]. Understanding these factors is vital to improve the health and QoL of parents, caregivers, and consequently, their children. Our study is the first to characterize the preoccupations, social aspects, and emotional features related to the QoL in parents or caregivers of Brazilian children or adolescents with CD by using a specific questionnaire. Therefore, it can help people manage their mental and physical wellbeing as well as the social limitations imposed by caring for children that are affected by this chronic disease [10,26].

Using an extensive literature review, and considering the suggestions made by experts such as nutritionists, pediatricians, and gastroenterologists with experience with CD, we constructed the first version of the questionnaire with 48 questions regarding QoL and 16 questions related to the characterization of the population including sociodemographic data (total of 64 questions). Once the QoL questions were created, we sent them to the judges to evaluate (Figure 1). Once the QoL questions were created, they were sent to the judges for evaluation (Figure 1). In this phase, the questionnaire related to the QoL was reduced from 48 to 33 questions (the 16 sociodemographic items were maintained). After the objective evaluation (comprised of semantic evaluation and content validation through the Delphi method), was completed 30 questions (10 in each domain) remained in addition to the 16 sociodemographic questions. The elimination of questions by the judges was advantageous, since a shorter questionnaire is more likely to be fully answered [43,44]. The CDPC-QoL took an average of six minutes to complete.

To assure the quality of the questionnaire, we performed a number of tests. The reliability test estimates the ability of the questionnaire to reproduce results, provided that no change in the outcome has taken place [37]. A confirmatory factor analysis was performed to verify the factor validity. We also measured the internal consistency (reliability) of the instrument (and its domains) by Cronbach’s alpha (Table 1), which was considered acceptable when the Cronbach’s alpha was ≥0.70 [45]. The internal consistency estimates the extent to which the included items of a score correlate with each other. We used Cronbach’s alpha coefficient to estimate this for the participants in all domains of the CWIS at the baseline [46]. The questionnaire also presented good internal consistency (α = 0.913; 95% CI: 0.892–0.932) for all domains (α > 0.8). The intraclass correlation coefficient (ICC) and the floor/ceiling effects were used to verify the reproducibility and the responsiveness of the questionnaire. The CDPC-QoL questionnaire presented good reproducibility (ICC = 0.88; *p* < 0.001) and good responsiveness (floor/ceiling effects ≤ 16.7%). In this sense, the CDPC-QoL presented good measures of reproducibility, indicating that similar results under consistent conditions are reproducible. In addition, the three domains of the questionnaire presented good fit in the confirmatory factor analysis (RMSEA < 0.05 and χ^2^ = 367.8; df = 344; *p* = 0.181).

After the semantic evaluation and content validation, the CDPC-QoL was applied to the parents or caregivers of celiac children or adolescents. Similar to the other studies of QoL [10,24,26,27], most of the respondents in this study were female (89%; *n =* 132), although the QoL did not differ with gender. We expected a larger number of female respondents since females tend to be more concerned about health, children’s health, and participate more often in health studies [8,18,47,48,49]. The study conducted by De Lorenzo et al. [8] with the parents of 33 celiac patients also had more responses from female participants (97%) compared to male participants in Brazil. In Brazil, the study conducted by Castilhos et al. [18] with celiac patients that were registered in the Southern Brazilian Celiac Association (ACELPAR) also showed a low rate of male participation (6.8%). The authors attributed it to the low participation rate of men in the Celiac Association meetings where the questionnaires were administered, and to the low concern of men on children’s health when compared to women [18]. However, it is important to mention that in our study questionnaire, we did not ask the participants about their affiliation to a CD association. Therefore, it is not possible to make the same assumption as the other authors [18]. A study that also examined the QoL of the parents of children with a chronic disease investigated the parents of children with type 1 diabetes (*n =* 134) which showed similar rates with the present study, showing that 90% of the participants as females [50]. Another study aimed to examine parental QoL (*n =* 463) in a sample of obese children participating in an inpatient program for treating obesity. Among the respondents, 77% were mothers; 8% were fathers and 15% were both fathers and mothers [51]. A study that evaluated the QoL among parents (*n =* 184) of children with autism disorder in the Arab world showed that 62% of the respondents were mothers and 38% were fathers [52].

Studies have shown a positive effect of family support to the QoL of celiac patients [9,19,53,54], supporting the idea that the marital status “with partner” has a positive effect on the celiac patient’s QoL. In our study, most of the respondents answered the marital status as “with a partner”, however, the marital status did not present a significant difference in QoL, unlike other studies that evaluated parents with children with another chronic disease that showed that single parents had a lower QoL [51,52]. Previous studies with celiac children’s parents did not evaluate the influence of marital status in QoL [8,16].

In our study, the higher the family income, the lower the preoccupations (higher score of the worries domain) (Table 2). We were expecting this result since, in Brazil, GF food is more costly when compared to gluten-containing products [11,55,56,57,58]. In addition, a study suggests that income modulates both health-seeking behavior and access to food and health care [59], which are both related to the higher QoL.

Our study showed that having another illness besides CD decreased the parents’ QoL. The association between CD and other aliments presents a challenge due to combined treatments, diet, and symptoms, which affect the patients’ QoL and consequently, the family’s sense of wellbeing [60]. Parents of children with health problems are at a higher risk for psychiatric disorders such as depression and anxiety [61]. Psychiatric disorders are common due to the psychological burden that parents of children with various chronic diseases suffer [62,63]. A study evaluated the anxiety and depression in 41,753 caregivers of CD individuals [62] where the authors showed that depression and anxiety levels were higher in parents of children with CD than in the controls (without CD) [62]. Corroborating these findings, we found that 18% (*n =* 27) of the participants make use of antidepressant or anti-anxiety medication. None of the studies asked about the use of antidepressant or anti-anxiety medication by parents of celiac children or adolescents, thus making it impossible to compare our findings in QoL. However, as mentioned previously, we did not find statistical differences in the QoL among the group that used or did not take medication. A study with 450 Brazilian celiac adults [9] showed that 17.5% of the celiac patients were using antidepressant or anti-anxiety medication, and those who did not take this kind of medication had a higher QoL, unlike our findings.

Our results showed that most of the parents (*n =* 114) mentioned that their child experienced CD symptoms with gluten consumption. However, regardless of the presence or absence of symptoms with gluten consumption, the QoL of the celiac parents was affected by their children’s disease. It is noteworthy that regardless of the age of their child, the QoL was also equally affected. Only 14% (*n =* 22) of the parents also presented with CD. However, the QoL did not differ from the group who were not diagnosed with CD.

This study presents some possible limitations. First, a relatively small sample size (*n =* 148) limited our ability to evaluate the impact of a child with CD on the parental quality of life. Further research should recruit a larger cohort of parents of children or adolescents to achieve higher statistical power. Our sample was larger than the other studies [8,16], but these findings can only provide a guide for future research. Second, our participation rate of male parents was 11%. Therefore, our results may not be generalizable. Future research should consider the inclusion of fathers to provide a broader understanding of the CD children/adolescents parents/caregivers’ QoL to increase generalizability. It is also important to highlight that the questionnaire was in Brazilian-Portuguese and transcultural studies should be carried out to analyze if this questionnaire may or may not be generalized to other Portuguese speaking countries. Brazilian-Portuguese and Portuguese from other countries such as Portugal are very distinct in terms of vocabulary, grammar, and spelling. Therefore, an adaptation should be conducted before applying it. In addition, the English version (Appendix A) has not been validated and is only a translation for a better understanding of the study. We mainly based our questionnaire on a previous CD-QoL questionnaire modified by a panel of professionals involved in the care of CD patients. Some questions seemed to be too negative; however, as researchers, we could not interfere in the panel of specialist’s decisions.

Another potential limitation is the inclusion criteria where we did not include the celiac patient’s biopsy proof of CD. However, since we used information based on a self-reported question, we asked them if a physician performed the diagnosis. Therefore, we assume the physician followed the ESPGHAN criteria to diagnose the children or adolescents [64].

This study also has important strengths. CDPC-QOL allowed us to evaluate the impact of a child’s disease on parents with a specific instrument. Moreover, this study provides a valuable insight into the potential mechanisms by which caring for a child with CD adversely impacts parental QoL. This information can help to design and implement effective and sustainable interventions to support parents who are experiencing excessive burden and stress, which may ultimately help prevent poor QoL outcomes.

## 5. Conclusions

The overall our results showed that the higher the family income, the lower the preoccupations (higher score of the worries domain). In addition, having another illness in addition to CD decreased the QoL (except in the worries domain). The other variables studied did not present a statistically significant impact on the QoL, which was shown to be low in all aspects.

The validation of a specific questionnaire of QoL is a valuable tool to evaluate the difficulties faced by parents or caregivers of children of this chronic illness. In this study, the validation of the CDPC-QoL was verified through semantic validity, content validity, construct reliability analysis, the questionnaire as a whole, and reproducibility analysis. Knowledge of the QoL is important to help implement effective strategies to improve the quality of life of celiac patients and in reducing the physical, emotional, and social burden on parents. We hope that the present study will help health professionals and governmental institutions in developing effective strategies to improve the QoL of celiac children and adolescents, and consequently, the QoL of Brazilian parents or caregivers.

## Figures and Tables

**Figure 1 nutrients-11-02302-f001:**
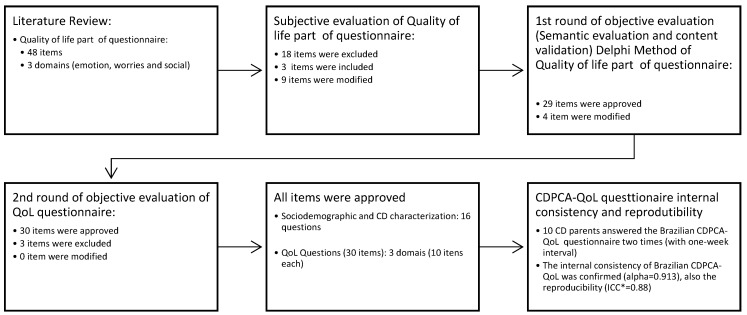
Process stages of the celiac disease parent/caregiver quality of life questions (CDPC-QOL), content validation, and semantic evaluation. * ICC, interclass correlation coefficient**.**

**Table 1 nutrients-11-02302-t001:** Internal consistency of the questionnaire by its domains.

Domain	N	Cronbach’s alpha (95% CI)	Floor Effect ^1^	Ceiling Effect ^2^
Emotional	10	0.830 (0.786–0.868)	0%	2.7%
Worries	10	0.821 (0.775–0.861)	0%	0%
Social	10	0.829 (0.785–0.867)	0%	0.7%
TOTAL	30	0.913 (0.892–0.932)	0%	0%

^1^ Floor effect was observed when the domain score = 10 or total score = 30 (worst results on the scale). ^2^ Ceiling effect was observed when the domain score = 50 or total score = 150 (best results on the scale).

**Table 2 nutrients-11-02302-t002:** Mean (M), standard deviation (SD), and *p*-value of comparison of the CDPCA-QoL questionnaire scores (and their domains) according to study variables.

	Domain			
Variable	Emotional M (SD)	Worries M (SD)	Social M (SD)	TOTAL M (SD)
**Overall (** ***n =*** ** 150)**	37.89 (7.37)	25.49 (8.35)	38.93 (7.76)	102.31 (20.06)
**Gender (** ***n =*** ** 148)**				
Female (*n =* 132)	37.64 (7.32) ^A^	25.40 (8.34) ^A^	38.65 (7.71) ^A^	101.70 (19.76)^A^
Male (*n =* 16)	40.06 (7.09) ^A^	25.81 (7.79) ^A^	41.06 (7.90) ^A^	106.94 (20.26)^A^
*p* *	0.212	0.852	0.241	0.319
**Age (** ***n =*** ** 147)**				
Up to 30 years (*n =* 10)	38.80 (7.64) ^A^	27.80 (7.61) ^A^	40.40 (6.35) ^A^	106.00 (20.53)^A^
31 a 40 (*n =* 64)	37.25 (6.94) ^A^	24.00 (7.54) ^A^	38.30 (6.89) ^A^	99.55 (18.18)^A^
41 a 50 (*n =* 60)	38.02 (8.04) ^A^	26.22 (9.18) ^A^	38.45 (8.70) ^A^	102.68 (22.18)^A^
Over 50 (*n =* 13)	38.77 (6.34) ^A^	25.39 (8.38) ^A^	41.54 (8.36) ^A^	106.85 (18.27)^A^
*p* **	0.893	0.335	0.492	0.538
**Child/adolescent CD Diagnostic Time (** ***n =*** ** 150)**				
Up to 11 months (*n =* 8)	35.88 (10.68) ^A^	26.75 (8.86) ^A^	39.50 (9.67) ^A^	102.13 (27.04) ^A^
12 a 23 months (*n =* 37)	37.16 (6.37) ^A^	24.49 (7.24) ^A^	38.46 (8.10) ^A^	100.11 (18.13) ^A^
24 a 35 months (*n =* 18)	37.22 (7.87) ^A^	21.39 (9.24) ^A^	37.22 (8.02) ^A^	95.83 (21.06) ^A^
36 a 59 months (*n =* 23)	36.87 (7.61) ^A^	27.00 (7.07) ^A^	38.83 (6.73) ^A^	102.70 (17.81) ^A^
Over 60 months (*n =* 64)	39.13 (7.27) ^A^	26.52 (8.86) ^A^	39.64 (7.74) ^A^	105.28 (20.71) ^A^
*p* **	0.507	0.148	0.815	0.447
**Education Level (** ***n =*** ** 150)**				
Elementary school (*n =* 15)	36.73 (8.44) ^A^	23.00 (7.93) ^A^	38.20 (8.61) ^A^	97.93 (22.30) ^A^
High School (*n =* 38)	38.47 (6.18) ^A^	22.82 (7.76)^ A^	40.68 (5.47) ^A^	101.97 (15.12) ^A^
College (*n =* 48)	38.06 (7.73) ^A^	25.69 (7.78) ^A,B^	38.77 (8.85) ^A^	102.52 (21.54) ^A^
Graduate or above (*n =* 49)	37.63 (7.69) ^A^	28.12 (8.82) ^B^	37.94 (7.87) ^A^	103.69 (21.60) ^A^
*p* **	0.877	0.016	0.412	0.813
**Marital status (** ***n =*** ** 149)**				
With partners (*n =* 135)	37.97 (7.23) ^A^	25.33 (8.16) ^A^	39.05 (7.69) ^A^	102.36 (19.72^ A^
Without partners (*n =* 14)	36.86 (9.05) ^A^	26.93 (10.52) ^A^	37.50 (8.79) ^A^	101.29 (24.51) ^A^
*p* *	0.593	0.500	0.479	0.851
**Occupation (** ***n =*** ** 148)**				
No (*n =* 37)	37.32 (8.09) ^A^	23.54 (8.60) ^A^	38.92 (8.81) ^A^	99.78 (22.35) ^A^
Yes (*n =* 111)	38.07 (7.20) ^A^	26.04 (8.23) ^A^	38.88 (7.49) ^A^	102.99 (19.44) ^A^
*p* *	0.597	0.116	0.981	0.404
**Age of child with CD (** ***n =*** ** 148)**				
Up to 5 years (*n =* 25)	37.40 (6.73) ^A^	22.84 (5.70) ^A^	38.16 (7.41) ^A^	98.40 (15.38) ^A^
6 a 10 (*n =* 59)	38.25 (7.45) ^A^	25.25 (8.65) ^A^	39.03 (7.71) ^A^	105.54 (20.32) ^A^
11 a 15 (*n =* 46)	36.83 (7.32) ^A^	26.02 (8.61) ^A^	38.43 (7.74) ^A^	101.28 (20.44) ^A^
Over 16 (*n =* 18)	40.22 (7.39) ^A^	28.50 (8.93) ^A^	40.94 (8.38) ^A^	109.67 (21.58) ^A^
*p* **	0.382	0.162	0.647	0.312
**Gender of child with CD (** ***n =*** ** 150)**				
Female (*n =* 93)	38.34 (7.40) ^A^	25.86 (8.54) ^A^	39.63 (7.27) ^A^	103.84 (19.76) ^A^
Male (*n =* 57)	37.16 (7.33) ^A^	24.88 (8.08) ^A^	37.77 (8.44) ^A^	99.81 (20.48) ^A^
*p* *	0.340	0.486	0.154	0.234
**Family income (** ***n =*** ** 148)**				
<1 MW *** (*n =* 14)	37.64 (8.46) ^A^	20.07 (7.07) ^A^	39.43 (8.81) ^A^	97.14 (21.57) ^A^
1 a 3 MW (*n =* 36)	36.25 (7.37) ^A^	22.25 (6.83) ^A,B^	38.11 (6.50) ^A^	96.61 (16.77) ^A^
3 a 6 MW (*n =* 32)	38.38 (7.50) ^A^	26.75 (8.18) ^A,B,C^	39.41 (8.26) ^A^	104.53 (21.14) ^A^
6 a 9 MW (*n =* 24)	36.25 (6.67) ^A^	25.96 (8.15) ^A,B,C^	37.83 (7.83) ^A^	100.04 (19.03) ^A^
9 a 12 MW (*n =* 16)	39.00 (8.41) ^A^	27.75 (7.82) ^B,C^	37.44 (9.23) ^A^	104.19 (21.97) ^A^
> 12 MW (*n =* 26)	40.46 (6.49) ^A^	29.65 (9.39) ^C^	40.73 (7.58) ^A^	110.85 (20.82) ^A^
*p* **	0.255	0.001	0.704	0.096
**Use of antidepressant or anti-anxiety medication (** ***n =*** ** 150)**				
No (*n =* 123)	38.41 (7.43) ^A^	25.67 (8.38) ^A^	39.14 (7.65) ^A^	103.22 (20.24) ^A^
Yes (*n =* 27)	35.52 (6.73) ^A^	24.67 (8.34) ^A^	37.96 (8.35) ^A^	98.15 (19.05) ^A^
*p* *	0.064	0.575	0.478	0.236
**Does the child have any other illness besides the CD? (** ***n =*** ** 148)**				
No (*n =* 107)	39.23 (6.71) ^A^	26.00 (8.22) ^A^	40.05 (7.12) ^A^	105.28 (18.57) ^A^
Yes (*n =* 41)	34.49 (8.08) ^B^	24.29 (8.76) ^A^	35.73 (8.64) ^B^	94.51 (22.25) ^B^
*p* *	0.000	0.269	0.002	0.003
**Do you have another relative with the CD? (** ***n =*** ** 150)**				
No (*n =* 99)	38.63 (6.86) ^A^	25.07 (8.02) ^A^	39.38 (7.22) ^A^	103.08 (18.81) ^A^
Yes (*n =* 51)	36.47 (8.16) ^A^	26.29 (9.00) ^A^	38.04 (8.73) ^A^	100.80 (22.42) ^A^
*p* *	0.090	0.397	0.316	0.512
**Are you celiac? (** ***n =*** ** 150)**				
No (*n =* 128)	37.86 (6.99) ^A^	24.98 (7.90) ^A^	38.88 (7.49) ^A^	101.72 (18.65) ^A^
Yes (*n =* 22)	38.09 (9.51) ^A^	28.45 (10.34) ^A^	39.18 (9.38) ^A^	105.73 (27.17) ^A^
*p* *	0.892	0.071	0.868	0.389
**Does your child experience discomfort or CD symptoms with gluten consumption? **(***n =***** 136)**				
No (*n =* 22)	35.14 (6.95) ^A^	25.91 (8.16) ^A^	38.64 (6.61) ^A^	99.68 (18.84) ^A^
Yes (*n =* 114)	37.74 (7.38) ^A^	25.30 (8.55) ^A^	38.41 (8.12) ^A^	101.45 (20.72) ^A^
*p* *	0.129	0.758	0.903	0.711

* Student *t*-test. ** ANOVA with Tukey posthoc test. **** MW, Minimum-wage: R$ 998.00 (about USD 250.00). Groups with the same letters do not significantly differ. On the same column, different letters represent statistical differences. Variables with n < 150 presented missing values.

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
