# Peer review of "Measuring Quality of Life in Parents or Caregivers of Children and Adolescents with Celiac Disease: Development and Content Validation of the Questionnaire"

_nutrients, 2019, doi:10.3390/nu11102302_

Round 1

Reviewer 1 Report

I think this paper is very original

It contains very useful scientific information

Your English is good

I have not suggestions or changes to do

Author Response

Thank you!

Reviewer 2 Report

Dear authors,

first of all thanks for this effort to create a QoL questionnaire for CD caregivers.

I do have some methodological concerns:

I would like to know who composed the "experts panel"? How many paediatricians, gastroenterologist, nutritionist? Were psychologist involved the creation of the questionnaire? If yes how? Were family organisation of parents with CD involved in the conception of the questionnaire? Reading the questionnaire gives a very negative impact on CD. Most of the question are actually proposed in a negative way and could rise a lot of doubt in CD caregivers.

Regarding discussion:

I feel that the discussion should be improved because literature proposed is very poor. Furthermore in the discussion I would like you to discuss upon which is the first questionnaire formulation based.

Regarding conclusions:

They are probably a bit to strong and a word of caution on the term validation should be add. Since Brazil have a population of 209 million people it is difficult to agree on a validation based on only 150 answer.

Author Response

Response to Reviewer 2 Comments

Dear reviewer,

Thank you for taking your time to review our article. Please let us know if you have any further concern _ we will be happy to address them.

I do have some methodological concerns:

I would like to know who composed the "experts panel"? How many paediatricians, gastroenterologist, nutritionist? Were psychologist involved the creation of the questionnaire? If yes how? Were family organisation of parents with CD involved in the conception of the questionnaire? Reading the questionnaire gives a very negative impact on CD. Most of the question are actually proposed in a negative way and could rise a lot of doubt in CD caregivers.

Answer to comment 1 –

I would like to know who composed the "experts panel"?  How many paediatricians, gastroenterologist, nutritionist?

The expert panel was proposed by the main author, her advisor and the head of the laboratory. Of those invited to participate in the panel four are pediatricians, three are pediatricians with specialization in gastroenterology, six gastroenterologists, one Nutritionist (with expertise in gluten-related disorders and quality of life), one dentist (with expertise in celiac disease and other gluten related disorders, and a psychologist who works with celiac patients and their families at the University Hospital CD outpatient clinic. We have no way on knowing is the psychologist was on the panel since the responses were anonymous.

Were psychologist involved the creation of the questionnaire?

If yes how?

A psychologist who worked with celiac patients at the celiac outpatient clinic in the past was consulted and asked for her input in the initial creation of the questionnaire.

Were family organisation of parents with CD involved in the conception of the questionnaire?

No – because we do not have a parents of CD patients (that we know of) in Brazil. 

Reading the questionnaire gives a very negative impact on CD. Most of the question are actually proposed in a negative way and could rise a lot of doubt in CD caregivers.

As we described in the manuscript, we used another questionnaires to compose this questionnaire after a vast literature review and after a panel of specialists suggest some modifications to compose the final version of the questionnaire.

Regarding discussion:

I feel that the discussion should be improved because literature proposed is very poor.

We will look into it

Furthermore in the discussion I would like you to discuss upon which is the first questionnaire formulation based.

I am not sure we understand – however, as previously mentioned – and as we described in the manuscript, we used another questionnaires to compose this questionnaire after a vast literature review and after a panel of specialists suggest some modifications to compose the final version of the questionnaire.

Regarding conclusions:

They are probably a bit to strong and a word of caution on the term validation should be add. Since Brazil have a population of 209 million people it is difficult to agree on a validation based on only 150 answer.

 In this article, the validation of the CDPC-QoL was verified through semantic validity, content validity, construct reliability analysis and the questionnaire as a whole and reproducibility analysis. For these last two cases whose verification is objective, the sample size is independent of the population size. And in this study one sample was sufficient, as shown by the confidence intervals (which were added in the text)

SPSS does not enforce Cronbach's Alpha Confidence Intervals, however Alpha is identical to ICC when we consider the consistency of a two-way mixed model (see below for IBM support). I advance this because he may question how the confidence intervals were calculated.

IBM SUPPORT

“Cronbach's alpha is identical to the intraclass correlation coefficient (ICC), when the ICC is calculated using either the two-way mixed consistency or two-way random consistency models. ICCs can be obtained through dialogs by clicking on the Statistics button under Analyze > Scale > Reliability Analysis, and checking the "Intraclass Correlation Coefficient" checkbox. A total of five ANOVA models are available through which the ICC may be calculated; as long as one chooses a two way consistency model (this is done through the two dropdowns labeled "Model" and "Type" just below the ICC checkbox), you will see in the output table titled "Intraclass Correlation Coefficient" a line labeled "Average Measures". The ICC on this line will be identical to Cronbach's Alpha, and a confidence interval is reported for the ICC; hence this is a CI for Alpha as well.

Reviewer 3 Report

Thank you very much for your work. The topic in interesting and scarcely studied. Despite the quality of the paper I would make some comments.

INTRODUCTION

1. The definition of Celiac Disease is correct but it could be more precise and more up to date. I suggest the use of  ESPGHAN definitions (2012) or that of Ludvingsson (2014). I believe that the introduction can be improved resuming CD Qol questionnaires and the dilemas CD patients and their parents face in their everyday life (Sverker 2007, 2009)

2. Line 49: We can consider spelt a variety of wheat as Kalmut© or triticale but authors have not made any reference to oats. We suggest to refer to all those cereals and their hybrids and probably oats (Comino, 2011)

3. Line 52: This idea should rely on a citation. 

MATERIAL AND METHODS

4. Line 104: Authors should describe here the members of the Delphi Panel. I miss psychologists as it is a QoL questionnaire.

5. Line 127: Modifies or modified?

6. Line 132: Is 10 CD parent´s responses a sample large enough to evaluate reproducibility? Please justify this number.

7. Sample size: Was a power analysis conducted?. Although the sample size is commented in the discussion section as a weakness of the study, some justification must be made about the suitability of the sample size for this number of items.

8. Sample composition: Do we know how many participants belonged to a Patients´association? Did authors considered being a member of a patient´association a relevant sociodemographic characteristic to be taken into account? This issue must be commented in the discussion section.

9. Were celiac patients biopsy proven? Were they recruited with a 

10. Regarding validity,  how was  construct, concurrent and discriminant validity assessed? 

11. Was a exploratory or confirmatory factor analysis carried out? Were the items grouped into the three proposed factors? What were the loads? What items belong to what domains? Please, improve the description of the questionnaire structure.

12. Did a factor analysis help to reduce the number of items?

13. How do we know we are measuring quality of life? Were other Qol questionnaires applied along this questionnaire? Were other questionnaires administered to assess convergent and discriminant validity? 

14. Line 145: Why were one year experienced parents/caregivers left out? Would they not be an interesting group to analyse their QoL and the impact of a diagnosis? In the same way, why were babies under 12 months left out?

15. Line 178: why a 80% of consensus is sufficient?. Please, justify.

16. Line 184: Revise redaction "in the questionnaire..."

17. Line 185: Can some questions be left in blank if they do not apply. Then the overall value could be below 30.

18. Line 187: Explain the three domains please, here or better in section 2.1.

19. Line 188: Can you offer a cut off point from which we can consider QoL low or high? Do other QoL questionnaires offer that cut off points? 

RESULTS

20. Please, report ceiling and floor effects.

21. Figure 1: Check for typo errors. (e.g. box 4: were excluded)

22. Table 2: Why do many n  add 148 or 150? Please, specify the treatment of missing values and change n value in table header.

23. Please indicate overall mean and SD for the sample (n=150) and for the 3 domains.

24. Line 197: Do not repeat internal consistency values, please.

APPENDIX

25. Please, unify the number of items between versions. Why does the English version have 31 items (item 20 is missing). I would also translate the sociodemographic questions (or rather eliminate them from the Brazilian version).

26. Including the questionnaire instruction could be useful to help with future transcultural adaptations.

27. Lines 931-932: ??? Should these lines be deleted??

28. Line 162: I suggest to describe the expert group in the "methods section"

29. Line 184: Linker?

30. Line 211: What was the size of the sample? 150? (line 211) 148? (line 338)

DISCUSSION 

31. Limitations: Address on the recruitment method: Most of them belonged to an association? Were they asked if they suffered from celiac disease and not from IBS or Wheat Allergy or NCGS? Were they biopsy proven? Were they asked if they were on a GFD? How strict were they?

32. Line 341: "Our sample was larger than other studies (please cite)".

CONCLUSION 

33. Line 358: What do authors mean by "Brazilian/Portuguese"? Transcultural studies should be carried out to analyse if this questionnaire is suitable for Portuguese population.

I would like to thank the authors on their effort for this paper. 

Author Response

Response to Reviewer 3 Comments

Thank your for taking your time to review our study. I hope we have answered them all to your satisfaction.

INTRODUCTION

The definition of Celiac Disease is correct but it could be more precise and more up to date. I suggest the use of ESPGHAN definitions (2012) or that of Ludvingsson (2014). I believe that the introduction can be improved resuming CD Qol questionnaires and the dilemas CD patients and their parents face in their everyday life (Sverker 2007, 2009)

Thank you, we will make these changes.

Line 49: We can consider spelt a variety of wheat as Kalmut© or triticale but authors have not made any reference to oats. We suggest to refer to all those cereals and their hybrids and probably oats (Comino, 2011)

We will make these changes

Line 52: This idea should rely on a citation. 

We will make these changes

MATERIAL AND METHODS

Line 104: Authors should describe here the members of the Delphi Panel. I miss psychologists as it is a QoL questionnaire.

Of those invited to participate in the panel four are pediatricians, three are paediatricians with specialization in gastroenterology, six gastroenterologists, and a psychologist and a dentist who works with celiac patients and their families at the University Hospital CD outpatient clinic, one nutritionist with expertise in gluten-related disorders. We have no way on knowing is the psychologist was on the panel since the responses were anonymous. Because we do not know who actually answer the questionnaire, we did not go into details. 

Line 127: Modifies or modified?

 Modified – corrected on text

Line 132: Is 10 CD parent´s responses a sample large enough to evaluate reproducibility? Please justify this number.

In this article, the validation of the CDPC-QoL was verified through semantic validity, content validity, construct reliability analysis, and the questionnaire as a whole and reproducibility analysis. For these last two cases whose verification is objective, the sample size is independent of the population size. Moreover, in this study, one sample was sufficient, as shown by the confidence intervals (which were added in the text).

Yes that number is sufficient. The p-values were included in the text, indicating that the ICC was significant.

Sample size: Was a power analysis conducted?. Although the sample size is commented in the discussion section as a weakness of the study, some justification must be made about the suitability of the sample size for this number of items.

The sample considered in this paper is sufficient to attest to the reliability of the constructs and the questionnaire as a whole, even for the questionnaire with this number of items. This statement is justified by Cronbach's alpha Confidence Intervals (which were included in the text). Note that all ranges showed lower limits greater than 0.7.

Sample composition: Do we know how many participants belonged to a Patients´association?

We do not know how many of the respondents belong to the CD association.

Did authors considered being a member of a patient´association.

Most of the authors belong to, volunteer for and help support the CD association.

a relevant sociodemographic characteristic to be taken into account? This issue must be commented in the discussion section.

We did not understand the sociodemographic question. We describe the sociodemographic characteristics of the population in table 2.

Were celiac patients biopsy proven? Were they recruited with a 

The protocol in Brazil for diagnosis of CD in children is for based purely on serological tests since those are 98 – 99% accurate. Unless there is a something out of the ordinary, we do no subject children to a biopsy.

Regarding validity, how was construct, concurrent and discriminant validity assessed? 

The constructs were constructed subjectively from the knowledge and expertise of the experts. The same happens with the discriminant validity of the questionnaire, which was verified by the semantic and content evaluation made by the experts (via Delphi method), and not by an objective analysis correlating the CDPCA-QoL with another Quality of Life Index.

Was a exploratory or confirmatory factor analysis carried out? Were the items grouped into the three proposed factors? What were the loads? What items belong to what domains? Please, improve the description of the questionnaire structure.

The constructs (and the equal weights of each item) were proposed and subjectively validated from the experts' opinion according to their opinions on the instrument's ability to measure what it intends to measure. The consistency of the constructs was subsequently validated (objectively) by Cronbach's Alpha.

Did a factor analysis help to reduce the number of items?

Factorial analysis is not appropriate here, as the questionnaire and its number of items have been defined and validated according to expert opinion.

How do we know we are measuring quality of life? Were other Qol questionnaires applied along this questionnaire? Were other questionnaires administered to assess convergent and discriminant validity? 

There are no other questionnaires that measure the quality of life of caregivers of children / adolescents with CD that we can compare to. Thus, an objective analysis of discriminant validity is not possible. However “the capacity of the instrument to accurately measure what it is intended to measure” was performed by the semantic and content analysis performed by the expert team.

By satisfying all aspects of validation analyzes and agreeing with expert opinion, this paper assumes, by definition, that CDPCA-QoL is a measure of due quality of caregivers of children / young people with CD.

Line 145: Why were one year experienced parents/caregivers left out? Would they not be an interesting group to analyse their QoL and the impact of a diagnosis? In the same way, why were babies under 12 months left out?

Several studies have shown that the first year of diagnosis of CD is very hard on patients and their families, because it is a year of adaptation. Therefore, we did not think it was a good idea to include them in the questionnaire.

Babies under 12 months were excluded because, CD normally presents itself only after the six months.  

Line 178: why a 80% of consensus is sufficient?. Please, justify.

The Delphi Method establishes a consensus over 70% - we decided to use over 80%. (Okoli & Pawlowski, 2004)

We will add the reference to the study.

Line 184: Revise redaction "in the questionnaire..."

I did not understand the correction.

Line 185: Can some questions be left in blank if they do not apply. Then the overall value could be below 30.

When a value was left blank one of two things happened. If it was only one item in the subsection, the it would be filled in with a total average for that subsection. If too many questions were left blank the a subsection the questionnaire was eliminates. 

Line 187: Explain the three domains please, here or better in section 2.1.

From lines 46 – 83 we talk about the possible physical, mental, emotional, and social aspects that may compromise activities such as vacations, trips, family events or restaurant outings for the patient, therefore, impacting the well-being, caused by CD in the daily lives of parents and caregivers.

Line 188: Can you offer a cut off point from which we can consider QoL low or high? Do other QoL questionnaires offer that cut off points? 

Since there is no other QoL questionnaire and the proposal is to use the general questionnaire and its domains, in this paper we do not propose a cutoff point for CDPC-QoL.

RESULTS

Please, report ceiling and floor effects.

The ceiling and floor effects were reported in Section 3.2.

Figure 1: Check for typo errors. (e.g. box 4: were excluded)

corrected

Table 2: Why do many n  add 148 or 150? Please, specify the treatment of missing values and change n value in table header.

The suggestion was addressed.

Please indicate overall mean and SD for the sample (n=150) and for the 3 domains.

The suggestion was addressed.

Line 197: Do not repeat internal consistency values, please.

Removed

APPENDIX

Please, unify the number of items between versions. Why does the English version have 31 items (item 20 is missing). I would also translate the sociodemographic questions (or rather eliminate them from the Brazilian version).

Done

Including the questionnaire instruction could be useful to help with future transcultural adaptations.

Lines 931-932: ??? Should these lines be deleted??

Why?

Line 162: I suggest to describe the expert group in the "methods section".

Done

Line 184: Linker?

Likert scale, corrected

Line 211: What was the size of the sample? 150? (line 211) 148? (line 338)

Ainda não fiz

DISCUSSION 

Limitations: Address on the recruitment method: Most of them belonged to an association? Were they asked if they suffered from celiac disease and not from IBS or Wheat Allergy or NCGS? Were they biopsy proven? Were they asked if they were on a GFD? How strict were they? Line 341: "Our sample was larger than other studies (please cite)".

We referenced the studies in the introduction - A study conducted in Brazil aimed to evaluate the QoL of parents (n = 63) of celiac children [6]using a generic questionnaire (WHOQoL-BREF).A study conducted in Turkey with mothers of celiac patients (n= 40) aimed to identify traumatic symptoms in mothers and to evaluate their QoL utilizing generic questionnaires "Post Traumatic Stress Disorder Checklist-Civilian Version (PCL-C)" and "Short Form-36 (SF-36)[12].To our knowledge, this is the first QoL questionnaire developed and validated to specifically explore the impact of CD on parents or caregivers of celiac children and adolescent.

CONCLUSION 

Line 358: What do authors mean by "Brazilian/Portuguese"? Transcultural studies should be carried out to analyse if this questionnaire is suitable for Portuguese population.

The questionnaire is not suitable for the Portuguese population. Brazilian Portuguese and Portuguese from other countries such as Portugal are very distinct in terms of vocabulary, grammar and spelling. Therefore, an adaptation should be conducted before applying it in other Portuguese speaking countries.

I would like to thank the authors on their effort for this paper. 

Thank you for your time and detailed review.

Round 2

Reviewer 2 Report

Dear authors, 

I still feel that discussion is not enough deep. 

In the previous revision I asked to add a part in the discussion regarding your choice of the questions. 

I totally understand that you based your choice on previous questionnaire, but how those questionnaire were commuted in your questionnaire is not specified in your paper. Since as a pediatric gastroenterologist I do feel that the question proposed to CD caregivers in your questionnaire are too negative I would like you to add a paragraph focusing on how questions were actually conceived and a paragraph of limitation on this specific topics. 

Thanks 

Author Response

Dear authors, 

I still feel that the discussion is not enough deep.

R: We inserted more information on discussion section.  

In the previous revision I asked to add a part in the discussion regarding your choice of the questions. 

R: We inserted information on this topic.

I totally understand that you based your choice on previous questionnaire, but how those questionnaire were commuted in your questionnaire is not specified in your paper. Since as a pediatric gastroenterologist I do feel that the question proposed to CD caregivers in your questionnaire are too negative I would like you to add a paragraph focusing on how questions were actually conceived and a paragraph of limitation on this specific topics. 

R: We inserted the information in methods and discussion section.

Reviewer 3 Report

Thank you for your effort in improving the paper. Below you can find some additional comments. 

Comments 26, 30 and 31 remain unanswered.

Thank you for taking your time to review our study. I hope we have answered them all to your satisfaction.

INTRODUCTION

The definition of Celiac Disease is correct but it could be more precise and more up to date. I suggest the use of ESPGHAN definitions (2012) or that of Ludvingsson (2014).

Thank you

I believe that the introduction can be improved resuming CD Qol questionnaires and the dilemas CD patients and their parents face in their everyday life (Sverker 2007, 2009)

Thank you, we will make these changes.

Thank you

Line 49: We can consider spelt a variety of wheat as Kalmut© or triticale but authors have not made any reference to oats. We suggest to refer to all those cereals and their hybrids and probably oats (Comino, 2011)

We will make these changes

Line 52: I still think unnecessary to mention spelt here but it´s ok.

Line 52: This idea should rely on a citation. 

We will make these changes

Thank you

MATERIAL AND METHODS

Line 104: Authors should describe here the members of the Delphi Panel. I miss psychologists as it is a QoL questionnaire.

Of those invited to participate in the panel four are pediatricians, three are paediatricians with specialization in gastroenterology, six gastroenterologists, and a psychologist and a dentist who works with celiac patients and their families at the University Hospital CD outpatient clinic, one nutritionist with expertise in gluten-related disorders. We have no way on knowing is the psychologist was on the panel since the responses were anonymous. Because we do not know who actually answer the questionnaire, we did not go into details. 

I do not quite understand the Delphi panel. In my opinion, the size was rather small as the 13 members could leave out some important field of expertise (those missing participants could be 3 paediatricians with specializaton in gastroenterology or 2 of them and a psychologist). Relying the validity of the questionnaire on this expertise requires a larger size with a more representative composition. I believe this aspect of the methodology is insufficient. It could have been improved including adolescents with cd or parents or members of the associations.

Line 127: Modifies or modified?

 Modified – corrected on text

Thank you

Line 132: Is 10 CD parent´s responses a sample large enough to evaluate reproducibility? Please justify this number.

In this article, the validation of the CDPC-QoL was verified through semantic validity, content validity, construct reliability analysis, and the questionnaire as a whole and reproducibility analysis. For these last two cases whose verification is objective, the sample size is independent of the population size. Moreover, in this study, one sample was sufficient, as shown by the confidence intervals (which were added in the text).

Yes that number is sufficient. The p-values were included in the text, indicating that the ICC was significant.

I do not agree. Some authors suggest a sample of 10 individuals per item (Kline 1998) for validation studies. A single Delphi panel with a size of 13 members cannot account for a complete validation process.

Sample size: Was a power analysis conducted?. Although the sample size is commented in the discussion section as a weakness of the study, some justification must be made about the suitability of the sample size for this number of items.

The sample considered in this paper is sufficient to attest to the reliability of the constructs and the questionnaire as a whole, even for the questionnaire with this number of items. This statement is justified by Cronbach's alpha Confidence Intervals (which were included in the text). Note that all ranges showed lower limits greater than 0.7.

I do not agree. Some authors suggest a sample of 10 individuals per item (Kline 1998) for validation studies. A single Delphi panel with a size of 13 members cannot account for a complete validation process.

Sample composition: Do we know how many participants belonged to a Patients´association?

We do not know how many of the respondents belong to the CD association.

Did authors considered being a member of a patient´association a relevant sociodemographic characteristic to be taken into account? This issue must be commented in the discussion section. This is a single question.

Most of the authors belong to, volunteer for and help support the CD association. This is irrelevant to me.

We did not understand the sociodemographic question. We describe the sociodemographic characteristics of the population in table 2. I was asking if participant´s affiliation to a CD associacion was asked in the questionnaire.

Perhaps I have not been clear enough and I may have been misunderstood. I do not refer to authors' affiliation but to the fact that If the majority of the sample belongs to a patients´association it may affect the representativeness of the sample and must be commented as a weakness and as a bias in the discussion

Were celiac patients biopsy proven? Were they recruited with a 

The protocol in Brazil for diagnosis of CD in children is for based purely on serological tests since those are 98 – 99% accurate (reference, please).Unless there is a something out of the ordinary, we do no subject children to a biopsy.

I only agree partially. ESPGHAN criteria (Husby; 2012) states "the clinical relevance of a positive anti-TG2 or anti-DGP result should be confirmed by histology, unless certain conditions are fulfilled that allow the option of omitting the confirmatory biopsies. If histology shows lesions that are consistent with CD (Marsh 2–3), then the diagnosis of CD is confirmed. If histology is normal (Marsh 0) or shows only increased IEL counts (>25 lymphocytes per 100 epithelial cells, Marsh 1), then further testing should be performed before establishing the diagnosis of CD"

ESPGHAN criteria also states that "in children and adolescents with signs or symptoms suggestive of CD and high anti-TG2 titers with levels >10 times ULN, the likelihood for villous atrophy (Marsh 3) is high. In this situation, the paediatric gastroenterologist may discuss with the parents and patient (as appropriate for age) the option of performing further laboratory testing (EMA, HLA) to make the diagnosis of CD without biopsies".

Authors should clarify if the inclusion criteria was by a self-reported question.

Additionally, must we assume that all the participants were in a GFD. Some patients with CD could not be in a strict GFD and we can assume some differences in QoL regarding this point. Was this asked or assumed?

Regarding validity, how was construct, concurrent and discriminant validity assessed? 

The constructs were constructed subjectively from the knowledge and expertise of the experts. The same happens with the discriminant validity of the questionnaire, which was verified by the semantic and content evaluation made by the experts (via Delphi method), and not by an objective analysis correlating the CDPCA-QoL with another Quality of Life Index.

I do not agree. Other questionnaires could have been administered along to study discriminant and convergent validity.

Was an exploratory or confirmatory factor analysis carried out? Were the items grouped into the three proposed factors? What were the loads? What items belong to what domains? Please, improve the description of the questionnaire structure.

The constructs (and the equal weights of each item) were proposed and subjectively validated from the experts' opinion according to their opinions on the instrument's ability to measure what it intends to measure. The consistency of the constructs was subsequently validated (objectively) by Cronbach's Alpha.

Although it is been designed according to expert opinion the statistical analysis could be improved with a confirmatory factor analysis. You can consider to add this factor analysis to your paper to see the contribution of each item, if the scale can be reduce and if we can confirm the 3 factor structure.

Did a factor analysis help to reduce the number of items?

Factorial analysis is not appropriate here, as the questionnaire and its number of items have been defined and validated according to expert opinion.

I do not agree. You could confirm with a factor analysis the work of the experts.

How do we know we are measuring quality of life? Were other Qol questionnaires applied along this questionnaire? Were other questionnaires administered to assess convergent and discriminant validity? 

There are no other questionnaires that measure the quality of life of caregivers of children / adolescents with CD that we can compare to. Thus, an objective analysis of discriminant validity is not possible. However “the capacity of the instrument to accurately measure what it is intended to measure” was performed by the semantic and content analysis performed by the expert team.

By satisfying all aspects of validation analyzes and agreeing with expert opinion, this paper assumes, by definition, that CDPCA-QoL is a measure of due quality of caregivers of children / young people with CD.

 Although it is probably the first questionnaire of its kind you could have administered along with other general QoL questionnaires like SF12 or SF36, for example. Other strategies can be consulted in Casellas (2013) or Dorn (2010) or Herrero (2014). discriminant and convergent validity could have also been calculated, using correlation coefficients with measures of depression, anxiety, self-esteem, positive and negative affect, optimism, and quality of life.

Line 145: Why were one year experienced parents/caregivers left out? Would they not be an interesting group to analyse their QoL and the impact of a diagnosis? In the same way, why were babies under 12 months left out?

Several studies have shown that the first year of diagnosis of CD is very hard on patients and their families, because it is a year of adaptation. Therefore, we did not think it was a good idea to include them in the questionnaire.

Babies under 12 months were excluded because, CD normally presents itself only after the six months.  

As you seem to agree, it could be an important moment to assess quality of life and study the differences.

Line 178: why a 80% of consensus is sufficient?. Please, justify.

The Delphi Method establishes a consensus over 70% - we decided to use over 80%. (Okoli & Pawlowski, 2004)

We will add the reference to the study

Thank you

Line 184: Revise redaction "in the questionnaire..."

In the questionnaire is composed of 30 items with answers on a five-point likert scale linker scale from 1 to 5. The value of the index was defined as the sum of the answers of each of these items.

I did not understand the correction.

It seem that "In" the questionnaire has to be deleted or something is missing (line 216)

Line 185: Can some questions be left in blank if they do not apply. Then the overall value could be below 30.

When a value was left blank one of two things happened. If it was only one item in the subsection, the it would be filled in with a total average for that subsection. If too many questions were left blank the a subsection the questionnaire was eliminates. 

According to your comment above, how many questionnaires were received and how many questionnaires were eliminated then?

Line 187: Explain the three domains please, here or better in section 2.1.

From lines 46 – 83 we talk about the possible physical, mental, emotional, and social aspects that may compromise activities such as vacations, trips, family events or restaurant outings for the patient, therefore, impacting the well-being, caused by CD in the daily lives of parents and caregivers.

I suggest to define each of the domains of the questionnaire. What's the difference between mental and emotional, for example?

As we do not have a factor analysis, I did not find which item belongs to which dimension. I think it useful to revise the content validity of the items.

Line 188: Can you offer a cut off point from which we can consider QoL low or high? Do other QoL questionnaires offer that cut off points? 

Since there is no other QoL questionnaire and the proposal is to use the general questionnaire and its domains, in this paper we do not propose a cutoff point for CDPC-QoL.

OK.

RESULTS

Please, report ceiling and floor effects.

The ceiling and floor effects were reported in Section 3.2.

I would not consider low a ceiling effect of >16% . It might be interesting to revise the following reference:

Terwee, C. B., Bot, S. D. M., de Boer, M. R., van der Windt, D. A. W. M., Knol, D. L., Dekker, J., … de Vet, H. C. W. (2007). Quality criteria were proposed for measurement properties of health status questionnaires. Journal of Clinical Epidemiology, 60(1), 34–42. https://doi.org/10.1016/j.jclinepi.2006.03.012

Thank you

Figure 1: Check for typo errors. (e.g. box 4: were excluded)

Corrected

Thank you

Table 2: Why do many n  add 148 or 150? Please, specify the treatment of missing values and change n value in table header.

The suggestion was addressed.

Thank you

Please indicate overall mean and SD for the sample (n=150) and for the 3 domains.

The suggestion was addressed.

Thank you

Line 197: Do not repeat internal consistency values, please.

Removed

Thank you

APPENDIX

Please, unify the number of items between versions. Why does the English version have 31 items (item 20 is missing). I would also translate the sociodemographic questions (or rather eliminate them from the Brazilian version).

Done

Including the questionnaire instruction could be useful to help with future transcultural adaptations.

This comment remains unanswered.

Lines 931-932: ??? Should these lines be deleted??

Why?

Because I believe they are part of the instructions for authors but not a part of the article. Please, check with Nutrients.

Line 162: I suggest to describe the expert group in the "methods section".

Done

Thank you

Line 184: Linker?

Likert scale, corrected

Thank you

Line 211: What was the size of the sample? 150? (line 211) 148? (line 338)

Ainda não fiz

Does this comment remain unanswered?

DISCUSSION 

Limitations: Address on the recruitment method: Most of them belonged to an association? Were they asked if they suffered from celiac disease and not from IBS or Wheat Allergy or NCGS? Were they biopsy proven? Were they asked if they were on a GFD? How strict were they?

Does this comment remain unanswered?

Line 341: "Our sample was larger than other studies (please cite)".

We referenced the studies in the introduction - A study conducted in Brazil aimed to evaluate the QoL of parents (n = 63) of celiac children [6]using a generic questionnaire (WHOQoL-BREF).A study conducted in Turkey with mothers of celiac patients (n= 40) aimed to identify traumatic symptoms in mothers and to evaluate their QoL utilizing generic questionnaires "Post Traumatic Stress Disorder Checklist-Civilian Version (PCL-C)" and "Short Form-36 (SF-36)[12].To our knowledge, this is the first QoL questionnaire developed and validated to specifically explore the impact of CD on parents or caregivers of celiac children and adolescent.

Please extend this answer addressing the rest of my comments.

CONCLUSION 

Line 358: What do authors mean by "Brazilian/Portuguese"? Transcultural studies should be carried out to analyse if this questionnaire is suitable for Portuguese population.

The questionnaire is not suitable for the Portuguese population. Brazilian Portuguese and Portuguese from other countries such as Portugal are very distinct in terms of vocabulary, grammar and spelling. Therefore, an adaptation should be conducted before applying it in other Portuguese speaking countries.

Ok, thank you. You could address this issue in the paper as well as the English version has not been validated and that it is only a translation for a better understanding of the study.

Thank you for your time and detailed review.

Thanks to you and your team for your effort. The paper has been improve in a great deal. The topic is really interesting but I think the design has important deficiencies (e.g. sample size, Delphi panel composition) and the statistical analysis should be improved in order to be published in a journal with this impact factor.

Please, check the reference below as a justification to many of my comments.

Terwee, C. B., Bot, S. D. M., de Boer, M. R., van der Windt, D. A. W. M., Knol, D. L., Dekker, J., … de Vet, H. C. W. (2007). Quality criteria were proposed for measurement properties of health status questionnaires. Journal of Clinical Epidemiology, 60(1), 34–42. https://doi.org/10.1016/j.jclinepi.2006.03.012

Author Response

Reviewer #3:

Thank you for your effort in improving the paper. Below you can find some additional comments. Comments 26, 30 and 31 remain unanswered.

R: We inserted more information to answer to these topics.

INTRODUCTION

Line 49: We can consider spelt a variety of wheat as Kalmut© or triticale but authors have not made any reference to oats. We suggest to refer to all those cereals and their hybrids and probably oats (Comino, 2011)

We will make these changes

Line 52: I still think unnecessary to mention spelt here but it´s ok.

R: Sorry, I believe that we misunderstood your comment. We removed the “spelt” from the manuscript.

MATERIAL AND METHODS

Line 104: Authors should describe here the members of the Delphi Panel. I miss psychologists as it is a QoL questionnaire.

Of those invited to participate in the panel four are pediatricians, three are paediatricians with specialization in gastroenterology, six gastroenterologists, and a psychologist and a dentist who works with celiac patients and their families at the University Hospital CD outpatient clinic, one nutritionist with expertise in gluten-related disorders. We have no way on knowing is the psychologist was on the panel since the responses were anonymous. Because we do not know who actually answer the questionnaire, we did not go into details. 

I do not quite understand the Delphi panel. In my opinion, the size was rather small as the 13 members could leave out some important field of expertise (those missing participants could be 3 paediatricians with specializaton in gastroenterology or 2 of them and a psychologist). Relying the validity of the questionnaire on this expertise requires a larger size with a more representative composition. I believe this aspect of the methodology is insufficient. It could have been improved including adolescents with cd or parents or members of the associations.

 R: We contact the specialists, and we obtained the correct information about the specialists' panel participants and we inserted in the manuscript. Therefore, it was composed by four pediatricians (three of them with specialization in gastroenterology); six gastroenterologists; one psychologist; one dietitian (with expertise in gluten-related disorders) and one dentist who works with celiac patients and their families at the University Hospital CD outpatient clinic. According to Pasquali [1], the judges need to present expertise in the area of the study, and a minimum of six individuals is necessary to compose the panel of the specialists. We invited the specialists to obtain a minimum of six, but we have more answers to compose our evaluation.

Line 132: Is 10 CD parent´s responses a sample large enough to evaluate reproducibility? Please justify this number.

In this article, the validation of the CDPC-QoL was verified through semantic validity, content validity, construct reliability analysis, and the questionnaire as a whole and reproducibility analysis. For these last two cases whose verification is objective, the sample size is independent of the population size. Moreover, in this study, one sample was sufficient, as shown by the confidence intervals (which were added in the text).

Yes that number is sufficient. The p-values were included in the text, indicating that the ICC was significant.

I do not agree. Some authors suggest a sample of 10 individuals per item (Kline 1998) for validation studies. A single Delphi panel with a size of 13 members cannot account for a complete validation process.

 R: In our manuscript, we used the Pasquali [1,2]recommendation to the CONTENT validation. According to Pasquali [1], for content validation, the judges need to present expertise in the area of the study, and a minimum of six individuals are necessary to compose the panel of the specialists. We invited the specialists to obtain a minimum of six, but we have more answers to compose our evaluation.The suggestion of 10 individuals per item is indicated to perform an exploratory factor analysis, which is a descriptive analysis. However, in this paper, the domains of the questionnaire were assembled by the experts. And the sample considered in the paper was sufficient to prove the experts' work (all results were statistically significant).

Sample size: Was a power analysis conducted?. Although the sample size is commented in the discussion section as a weakness of the study, some justification must be made about the suitability of the sample size for this number of items.

 The sample considered in this paper is sufficient to attest to the reliability of the constructs and the questionnaire as a whole, even for the questionnaire with this number of items. This statement is justified by Cronbach's alpha Confidence Intervals (which were included in the text). Note that all ranges showed lower limits greater than 0.7.

I do not agree. Some authors suggest a sample of 10 individuals per item (Kline 1998) for validation studies. A single Delphi panel with a size of 13 members cannot account for a complete validation process.

 R: As we mention in the previous question, we used the content validation by Delphy technic using a panel of specialists, as recommended by Pasquali[1,2]. We inserted more information about this in the manuscript.

Sample composition: Do we know how many participants belonged to a Patients´association?

We do not know how many of the respondents belong to the CD association.

Did authors considered being a member of a patient´association a relevant sociodemographic characteristic to be taken into account? This issue must be commented in the discussion section. 

Most of the authors belong to, volunteer for and help support the CD association. This is irrelevant to me.

R: Sorry for our mistake. Now we understood the question and inserted the information in the manuscript.

We did not understand the sociodemographic question. We describe the sociodemographic characteristics of the population in table 2. I was asking if participant´s affiliation to a CD associacion was asked in the questionnaire.

Perhaps I have not been clear enough and I may have been misunderstood. I do not refer to authors' affiliation but to the fact that If the majority of the sample belongs to a patients´association it may affect the representativeness of the sample and must be commented as a weakness and as a bias in the discussion

 R: Sorry, previously, we did not understand that. Now, we inserted the information on the manuscript.

Were celiac patients biopsy proven?

I only agree partially. ESPGHAN criteria (Husby; 2012) states "the clinical relevance of a positive anti-TG2 or anti-DGP result should be confirmed by histology, unless certain conditions are fulfilled that allow the option of omitting the confirmatory biopsies. If histology shows lesions that are consistent with CD (Marsh 2–3), then the diagnosis of CD is confirmed. If histology is normal (Marsh 0) or shows only increased IEL counts (>25 lymphocytes per 100 epithelial cells, Marsh 1), then further testing should be performed before establishing the diagnosis of CD"

ESPGHAN criteria also states that "in children and adolescents with signs or symptoms suggestive of CD and high anti-TG2 titers with levels >10 times ULN, the likelihood for villous atrophy (Marsh 3) is high. In this situation, the paediatric gastroenterologist may discuss with the parents and patient (as appropriate for age) the option of performing further laboratory testing (EMA, HLA) to make the diagnosis of CD without biopsies".

Authors should clarify if the inclusion criteria was by a self-reported question.

Additionally, must we assume that all the participants were in a GFD. Some patients with CD could not be in a strict GFD and we can assume some differences in QoL regarding this point. Was this asked or assumed?

 R: We inserted the information in the methods and discussion section.

Regarding validity, how was construct, concurrent and discriminant validity assessed? 

The constructs were constructed subjectively from the knowledge and expertise of the experts. The same happens with the discriminant validity of the questionnaire, which was verified by the semantic and content evaluation made by the experts (via Delphi method), and not by an objective analysis correlating the CDPCA-QoL with another Quality of Life Index.

I do not agree. Other questionnaires could have been administered along to study discriminant and convergent validity.

R: The convergent validity was vefiried by confirmatory factor analysis and its results was added in the manuscript. Discriminant validation was done by the experts..

Was an exploratory or confirmatory factor analysis carried out? Were the items grouped into the three proposed factors? What were the loads? What items belong to what domains? Please, improve the description of the questionnaire structure.

The constructs (and the equal weights of each item) were proposed and subjectively validated from the experts' opinion according to their opinions on the instrument's ability to measure what it intends to measure. The consistency of the constructs was subsequently validated (objectively) by Cronbach's Alpha.

Although it is been designed according to expert opinion the statistical analysis could be improved with a confirmatory factor analysis. You can consider to add this factor analysis to your paper to see the contribution of each item, if the scale can be reduce and if we can confirm the 3 factor structure.

R: Confirmatory factor analysis was performed, and its result added to the manuscript. The domains do not need to be reduced and/or the values of the items modified, as confirmatory factor analysis confirmed the sturdiness of the three domains of the questionnaires (D1: Q1-10; D2: Q11-10 and D3: Q21-30, all with the same weight).

Did a factor analysis help to reduce the number of items?

Factorial analysis is not appropriate here, as the questionnaire and its number of items have been defined and validated according to expert opinion.

I do not agree. You could confirm with a factor analysis the work of the experts.

R: Ok, we performed the confirmatory factor analysis, and its result added to the manuscript.

How do we know we are measuring quality of life? Were other Qol questionnaires applied along this questionnaire? Were other questionnaires administered to assess convergent and discriminant validity? 

There are no other questionnaires that measure the quality of life of caregivers of children / adolescents with CD that we can compare to. Thus, an objective analysis of discriminant validity is not possible. However “the capacity of the instrument to accurately measure what it is intended to measure” was performed by the semantic and content analysis performed by the expert team.

By satisfying all aspects of validation analyzes and agreeing with expert opinion, this paper assumes, by definition, that CDPCA-QoL is a measure of due quality of caregivers of children / young people with CD.

 Although it is probably the first questionnaire of its kind you could have administered along with other general QoL questionnaires like SF12 or SF36, for example. Other strategies can be consulted in Casellas (2013) or Dorn (2010) or Herrero (2014). discriminant and convergent validity could have also been calculated, using correlation coefficients with measures of depression, anxiety, self-esteem, positive and negative affect, optimism, and quality of life.

 R: The confirmatory factor analysis was performed, and that the experts' work was confirmed by confirmatory factor analysis.

Line 145: Why were one year experienced parents/caregivers left out? Would they not be an interesting group to analyse their QoL and the impact of a diagnosis? In the same way, why were babies under 12 months left out?

Several studies have shown that the first year of diagnosis of CD is very hard on patients and their families, because it is a year of adaptation. Therefore, we did not think it was a good idea to include them in the questionnaire.

Babies under 12 months were excluded because, CD normally presents itself only after the six months.  

As you seem to agree, it could be an important moment to assess quality of life and study the differences.

R: Sorry, we believe that our previous response was not so clear. We wanted to measure the QoL, after the adaptation period of at least 12 months – since it is well documented that the first years is related to low QoL.

Additionally, we inserted the information in the manuscript: “The cut point for “under 12 months” was used based on the inclusion criteria, in which the children should haveless than a year of diagnosis. Since in Brazil, the Brazilian Society of Pediatrics[3]recommends the exclusive breastfeeding until six months of age, and the introduction of gluten after six months, children under 12 months cannot haveless than a year of CD diagnosis”.

Line 184: Revise redaction "in the questionnaire..."

It seem that "In" the questionnaire has to be deleted or something is missing (line 216)

R: We deleted the word “in”. Sorry for our mistake.

Line 185: Can some questions be left in blank if they do not apply. Then the overall value could be below 30.

When a value was left blank one of two things happened. If it was only one item in the subsection, the it would be filled in with a total average for that subsection. If too many questions were left blank the a subsection the questionnaire was eliminates. 

According to your comment above, how many questionnaires were received and how many questionnaires were eliminated then?

R:  In this study, all individuals who agreed to complete the questionnaire did it completely considering the domains. Some missing were observed, but only in sociodemographic variables. Therefore, to construct the table 1, in which we evaluated the domain and compared the responses to sociodemographic variables, we used the responses to each

Line 187: Explain the three domains please, here or better in section 2.1.

R: We explained the three domains in the methods section.

From lines 46 – 83 we talk about the possible physical, mental, emotional, and social aspects that may compromise activities such as vacations, trips, family events or restaurant outings for the patient, therefore, impacting the well-being, caused by CD in the daily lives of parents and caregivers.

I suggest to define each of the domains of the questionnaire. What's the difference between mental and emotional, for example?

R: We inserted the information in the method section.

As we do not have a factor analysis, I did not find which item belongs to which dimension. I think it useful to revise the content validity of the items.

R: we performed the confirmatory factor analysis, and its result added to the manuscript.

RESULTS

Please, report ceiling and floor effects.

The ceiling and floor effects were reported in Section 3.2.

I would not consider low a ceiling effect of >16% . It might be interesting to revise the following reference:

Terwee, C. B., Bot, S. D. M., de Boer, M. R., van der Windt, D. A. W. M., Knol, D. L., Dekker, J., … de Vet, H. C. W. (2007). Quality criteria were proposed for measurement properties of health status questionnaires. Journal of Clinical Epidemiology60(1), 34–42. https://doi.org/10.1016/j.jclinepi.2006.03.012

 R: Thank you! We used the reference that you recommended and modified the table as your suggestion.

Including the questionnaire instruction could be useful to help with future transcultural adaptations.

This comment remains unanswered.

R: We inserted more information in method and discussion section. We hope that it is better explained now.

  Lines 931-932: ??? Should these lines be deleted??

Because I believe they are part of the instructions for authors but not a part of the article. Please, check with Nutrients.

R: We removed the information. Sorry, but in the previous version we saw the wrong lines, and we did not perceive that we were wrong.

Line 211: What was the size of the sample? 150? (line 211) 148? (line 338)

Does this comment remain unanswered?

R: we explained it in the results section: “It is also important to highlight that the variables that present n <150, presented missing, in which the participant did not mark any response in the item of the questionnaire”.

DISCUSSION 

Limitations: Address on the recruitment method: Most of them belonged to an association? Were they asked if they suffered from celiac disease and not from IBS or Wheat Allergy or NCGS? Were they biopsy proven? Were they asked if they were on a GFD? How strict were they?

 Does this comment remain unanswered?

 R: We used the Brazilian Celiac Associations (ACELBRA) to help us to divulge the research among their associates. However, we also used social media like Facebook, Instagram, emails from our patients to reach the participants. Therefore, since we did not ask a question about their association with the ACELBRAs, it is not possible to allege that they belong to an association.

The participants were asked about their children diagnosis (if it was performed by a physician) – which were a inclusion criteria, as we included in the manuscript. We also inserted in the manuscript that we did not use their children biopsy-proven, only the self-report of the participants.

Line 341: "Our sample was larger than other studies (please cite)".

R: We inserted the references (previously mentioned in the introduction section).

We referenced the studies in the introduction - A study conducted in Brazil aimed to evaluate the QoL of parents (n = 63) of celiac children [6]using a generic questionnaire (WHOQoL-BREF).A study conducted in Turkey with mothers of celiac patients (n= 40) aimed to identify traumatic symptoms in mothers and to evaluate their QoL utilizing generic questionnaires "Post Traumatic Stress Disorder Checklist-Civilian Version (PCL-C)" and "Short Form-36 (SF-36)[12].To our knowledge, this is the first QoL questionnaire developed and validated to specifically explore the impact of CD on parents or caregivers of celiac children and adolescent.

Please extend this answer addressing the rest of my comments.

CONCLUSION 

Line 358: What do authors mean by "Brazilian/Portuguese"? Transcultural studies should be carried out to analyse if this questionnaire is suitable for Portuguese population.

The questionnaire is not suitable for the Portuguese population. Brazilian Portuguese and Portuguese from other countries such as Portugal are very distinct in terms of vocabulary, grammar and spelling. Therefore, an adaptation should be conducted before applying it in other Portuguese speaking countries.

Ok, thank you. You could address this issue in the paper as well as the English version has not been validated and that it is only a translation for a better understanding of the study.

R: We inserted the information at the end of the discussion section.

Thank you for your time and detailed review.

Thanks to you and your team for your effort. The paper has been improve in a great deal. The topic is really interesting but I think the design has important deficiencies (e.g. sample size, Delphi panel composition) and the statistical analysis should be improved in order to be published in a journal with this impact factor.

Please, check the reference below as a justification to many of my comments.

Terwee, C. B., Bot, S. D. M., de Boer, M. R., van der Windt, D. A. W. M., Knol, D. L., Dekker, J., … de Vet, H. C. W. (2007). Quality criteria were proposed for measurement properties of health status questionnaires. Journal of Clinical Epidemiology60(1), 34–42. https://doi.org/10.1016/j.jclinepi.2006.03.012

R: Thank you for your contribution. We read the manuscript and it was useful to improve our work.

Thank you for the opportunity!

References:

Pasquali, L. Psicometria. Rev. da Esc. Enferm. da USP2009, 43, 992–999, doi:10.1590/S0080-62342009000500002. Pasquali, L. Psicometria: Teoria dos testes na psicologia e na educação - Luiz Pasquali - Google Livros; 1st ed.; Vozes: Brasilia, 2017; ISBN 8532656129, 9788532656124. Sociedade Brasileira de Pediatria Manual de orientação do departamento de nutrologia: alimentação do lactente ao adolescente, alimentação na escola, alimentação saudável e vínculo mãe-fi lho, alimentação saudável e prevenção de doenças, segurança alimentar; 3rd ed.; SBP: Rio de Janeiro, 2012; ISBN 978-85-88520-22-6.

Round 3

Reviewer 2 Report

Thank you for the modifications. 

Author Response

This is part of the answer for review 3. 

Reviewer 3 Report

Thank you very much for your effort in improving the manuscript.

I still think that it could be improved showing the factor loading matrix of the confirmatory analysis.

Regards,

Author Response

3ndReview of manuscript ID nutrients-563440 with the title"Measuring Quality of Life in Parents or Caregivers of children and adolescents with Celiac Disease: Development and Content Validation of the Questionnaire”

Reviewer #3:

Thank you very much for your effort in improving the manuscript. I still think that it could be improved showing the factor loading matrix of the confirmatory analysis.

R: In the confirmatory factor analysis, the structure that was tested was the same adopted for the calculation of the questionnaire scores (all items have the same weight). Therefore, the structure that must be confirmed by the Confirmatory Factor Analysis does not have the loading factors free (regression weights). In fact, they must be fixed with equal values because, if the loading factors were different, it would mean that the items of the questionnaires should have different weights to obtain the score. Only the covariances of the exogenous variables are free parameters in this model.

In this context, the presentation of the loadings factor is unnecessary, since it is already known that the confirmatory factor analysis should be performed by setting the regression coefficients. In fact, the questionnaire structure is validated by verifying that the covariance structure of the questionnaire structure (Default model) does not differ significantly from the covariance structure of the observed data. In our study, this divergence was measured by the Chi-square discrepancy test (CMIN), which did not reject the adopted structure (CMIN = 367.80, p = 0.181).]

 If our questionnaire presented items with different weights, an adjustment would be necessary. However, in our study, it is of interest that the questionnaire has equal weights for each item. Even so, the data obtained were not able to reject statistically the structure adopted for the questionnaire.

We attached some documents to facilitate the comprehension of our explanation (questionnaire structure; model fit summary; Estimate tables -  please see all regression weights fixed equal to 1).

Please see Factor Loading in attached doc.
